# Against *Clostridioides difficile* Infection: An Update on Vaccine Development

**DOI:** 10.3390/toxins17050222

**Published:** 2025-05-01

**Authors:** Jingyao Wang, Qianquan Ma, Songhai Tian

**Affiliations:** 1State Key Laboratory of Natural and Biomimetic Drugs, Department of Molecular and Cellular Pharmacology, School of Pharmaceutical Sciences, Peking University, 38 Xueyuan Road, Beijing 100191, China; 2Department of Neurosurgery, Peking University Third Hospital, Beijing 100191, China

**Keywords:** *Clostridioides difficile*, CDI, TcdA, TcdB, vaccine development

## Abstract

*Clostridioides difficile* (*C. difficile*) is a major pathogen responsible for antibiotic-associated diarrhea, frequently observed in hospital settings. Due to the widespread use of antibiotics, the incidence and severity of *C. difficile* infection (CDI) are rising across the world. CDI is primarily driven by two homologous protein exotoxins, toxin A (TcdA) and toxin B (TcdB). Other putative virulence factors include binary toxin CDT, surface layer proteins, phosphorylated polysaccharides, and spore coat proteins. These *C. difficile* virulence factors are potential targets for vaccine development. Although several *C. difficile* vaccines have entered clinical trials, there is currently no approved vaccine on the market. This review outlines the intoxication mechanism during CDI, emphasizing the potential antigens that can be used for vaccine development. We aim to provide a comprehensive overview of the current status of research and development of *C. difficile* vaccines.

## 1. Introduction

*Clostridioides difficile* (*C. difficile*), a Gram-positive, obligate anaerobic bacterium, is widely distributed in natural environments and the gastrointestinal tracts of domesticated animals and humans [1,2]. As the primary pathogen responsible for hospital-acquired diarrhea [3], *C. difficile* infections (CDI) develop when the human gut microbiota is disrupted, allowing *C. difficile* to proliferate excessively and release toxins. This pathogenic process leads to mucosal congestion, edema, and exudation, with diarrhea serving as the primary clinical manifestation [4,5,6]. CDI presents a significant public health challenge, and the US Centers for Disease Control and Prevention (CDC) has designated it as an urgent threat [7,8,9]. *C. difficile* causes nearly 500,000 infections and 29,000 associated deaths annually in the United States [10,11]. Additionally, the recurrence rate of CDI in cancer patients ranges from 13% to 35% in most studies [12]. Currently, the treatment of CDI primarily involves antimicrobial agents such as metronidazole, vancomycin, and fidaxomicin [13,14,15]. Tigecycline might be a potential therapeutic option for severe CDI [13], and rifaximin might effectively prevent CDI recurrence [16]. Unfortunately, these antimicrobial agents are not always effective in preventing these consequences, approximately one-third of treated patients still experience recurrence [11]. Fecal microbiota transplantation (FMT) has high clinical efficacy in treating recurrent *C. difficile* infections. However, the mechanisms are incompletely understood, and there are infection risks [17,18]. Due to the severe consequences of CDI, using vaccines to prevent CDI is a promising method. In this review, the pathogenic mechanisms of *C. difficile* toxins, the recent advances in vaccine development, and innovative approaches for future CDI prevention are discussed.

## 2. Biology of *C. difficile* Toxins

Pathogenic *C. difficile* strains harbor a 19.6 kb pathogenicity locus (PaLoc), which includes six genes: *tcdA*, *tcdB*, *tcdR*, *tcdE*, *tcdL,* and *tcdC* (which encode TcdA, TcdB, TcdR, TcdE, TcdL, and TcdC, respectively) [19,20,21] (Figure 1a). TcdA and TcdB are two protein exotoxins that directly regulate many physiological events within tissues and lead to disease [22]. TcdR is a DNA binding protein that positively regulates the expression of TcdA and TcdB [23,24]; TcdC is a negative regulator of the expression of TcdA and TcdB and acts as a membrane-bound anti-σ-factor [25,26]; TcdE facilitates the release of TcdA and TcdB from the *C. difficile* cells [27]. TcdL is a short fragment of an endolysin remnant of a phage holin/endolysin pair which binds with TcdB to mediate its transport [28]. Beyond the PaLoc, binary toxin (also called *C. difficile* transferase, CDT), a third exotoxin comprising CDTa and CDTb subunits, is produced by some hypervirulent *C. difficile* strains (e.g., NAP1/BI/027) [29]. These subunits are independently synthesized and secreted but assemble into functional holotoxin complexes spontaneously [25,30].

Structurally, TcdA and TcdB share four conserved functional domains (Figure 1b,c): an N-terminal glucosyltransferase domain (GTD), a cysteine protease domain (CPD), a delivery and receptor-binding domain (DRBD), and a C-terminal domain called combined repetitive oligopeptides (CROPs) [22,25]. As key virulence determinants, strains of *C. difficile* that produce TcdA and TcdB drive CDI pathogenesis [22]. The administration of toxin-neutralizing antibodies has been found to protect against CDI, indicating that TcdA and TcdB are primary target antigens in *C. difficile* vaccine development [31,32,33].

In recent years, a growing number of *C. difficile* strains have been isolated and analyzed [33,34,35,36]. According to a publicly available database, DiffBase (https://diffbase.uwaterloo.ca/, accessed on 1 October 2024) [37], TcdA sequences are clustered into seven subtypes (TcdA1-A7), and TcdB sequences are clustered into 12 subtypes (TcdB1-B12), respectively. The reference strain VPI10463 has been designated to express the archetype of toxins TcdA1 and TcdB1 [38]. In addition, each unique toxin variant within a subtype can be further subdivided (e.g., TcdB1.1). These sequence variations may alter toxin structural details, potentially affecting functional properties and antigenic profiles [39]. Despite these functional variations, the toxin variants maintain high sequence similarity—a likely consequence of their large size. A sequence alignment analysis has revealed that the TcdA subtypes A1–A6 share > 97% identity; a distinct TcdA7 subtype maintains ~85% identity with others; while the TcdB subtypes show greater dispersion but still maintain > 85.3% identity [36,37]. This high sequence conservativity suggests that numerous surface epitopes may be suitable for developing unique vaccines that confer protection across multiple subtypes. On the other hand, epidemiological data from clinical isolates indicate that three dominant subtypes account for most infections: TcdA1, TcdB1, and TcdB2 [37]. This prevalence pattern strongly supports focusing vaccine development efforts on these three toxin subtypes.

The intoxication mechanisms of TcdA and TcdB are illustrated in Figure 2 [21,40]. Toxins first bind to their specific receptors on the cell surface through their DRBD and CROPs domains. Glycoprotein 96 (gp96), low-density lipoprotein receptor (LDLR), and sulfated glycosaminoglycans (sGAGs) have been reported to facilitate TcdA cellular binding and entry [41,42,43]. TcdB has been reported to bind with Nectin 3 (also termed “poliovirus receptor-like protein 3”, PVRL3), frizzled receptor 1/2/7 (FZD1/2/7), low-density lipoprotein receptor-related protein 1 (LRP1), tissue factor pathway inhibitor (TFPI), and chondroitin sulfate proteoglycan 4 (CSPG4) [33,44,45,46,47]. Upon receptor binding, both toxins utilize endocytic pathways for internalization. The pH reduction in endosomes (caused by the influx of protons), triggers conformational changes of DRDB domains, facilitating the translocation of the CPD and the GTD across the endosomal membrane to the cytoplasm [48]. The CPD is then activated by inositol hexakisphosphate (InsP_6_), leading to the release of the GTD through autoproteolytic cleavage [49,50,51,52]. Although both toxins share a similar mechanism, TcdB is more susceptible to InsP_6_-induced self-cleavage than TcdA [51]. The free GTD targets and catalyzes the glucosylation of small guanosine triphosphatases (GTPases, such as Rho and/or Ras family members) using UDP-glucose as a sugar donor, and thereby inhibiting their functions, including the organization of the actin cytoskeleton [21,53,54]. Finally, the inactivation of small GTPases culminates in actin condensation, characteristic cell rounding, and eventual cell death [55].

## 3. Strategies for Vaccine Development

Vaccination remains one of the most effective strategies for preventing human infectious diseases [56]. Traditional vaccines induce immune responses to generate pathogen-specific antibodies or effector T-cells [57]. However, imitations such as complex formulations and safety concerns have driven the development of second-generation vaccines. These rely on genetic and protein engineering technologies to induce immunity through precisely defined antigens instead of whole-pathogen components, thereby minimizing risks while improving vaccine efficacy [58]. The third-generation vaccines are mRNA vaccines, which employ RNA as their core material, mark the most recent innovation, and significantly revolutionize the development of vaccines [59] (Figure 3). Next, strategies for vaccine design will be discussed in details.

### 3.1. Traditional Vaccines

Live attenuated vaccines (LAVs) are produced by reducing the virulence of pathogens while maintaining their viability. Well-known examples include the Bacillus Calmette–Guérin (BGG) vaccine (against tuberculosis) and the cholerae vaccine [60]. Bacterial attenuation typically occurs through serial passage, with most LAVs demonstrating over 90% efficacy and a multiyear protective duration [61]. The notable advantages of LAVs are that the quantity, nature, and location of antigens produced by the immune reaction are similar to those encountered during natural infection. Upon administration, LAVs trigger robust humoral and cellular immunity, generating pathogen-specific antibodies (Figure 3a). A single dose of administration often provides adequate protection. However, residual risks of viral reversion to virulence persist [58]. Inactivated vaccines (IVs) are developed by completely killing pathogens using heat, chemicals, or radiation, resulting in non-replicative antigens. Examples include the pertussis vaccine [62] (Figure 3b). Since IVs cannot reproduce in the host, their safety is improved. However, their immunogenicity is also weaker than LAVs. To compensate for this weakness, adjuvants are often added, and multiple immunizations are required to achieve long-lasting protection [63]. IVs predominantly stimulate humoral immune responses and are associated with mild side effects [58]. Toxoid vaccines (TVs) are of great significance for diseases caused by toxins, such as tetanus and diphtheria [64,65]. These formulations chemically (e.g., formalin) or thermally inactivate bacterial toxins while retaining immunogenic epitopes to elicit the immune responses [66] (Figure 3c). TVs primarily drive humoral immunity with limited cellular response, often requiring adjuvants and repeated administration for durable protection [58].

### 3.2. Second-Generation Vaccines

Advances in protein engineering and deeper insights into toxin pathogenesis have expanded the applications of subunit vaccines (SVs) [67]. SVs utilize targeted antigen fragments rather than whole pathogens, eliminating disease transmission risks. Their production involves inserting genetic constructs into chassis cells (e.g., bacterial, yeast, or mammalian systems) followed by the recombinant purification of antigens (Figure 3d). SV formulations benefit from streamlined composition and simplified quality control. Notably, the non-cellular pertussis vaccine and hepatitis B surface antigen (HBsAg) vaccine have demonstrated high immunogenicity at low doses. However, a single subunit antigen often inadequately stimulates immunity, necessitating combinations with potent adjuvants to activate both humoral and cellular immune pathways [68,69]. Recombinant toxin vaccines (RTVs) are engineered after the toxin proteins are expressed under recombinant conditions (Figure 3e). Key mutations are introduced into the toxins to block their native functions. Upon administration, RTVs provoke dual humoral and cellular responses. In addition, large-scale culture technology can reduce costs, and the ingredients are uniform and stable, offering better safety. Despite these advantages, RTVs typically exhibit limited immunogenicity, frequently requiring adjuvant support and booster doses to sustain protective immunity [70].

### 3.3. Third-Generation Vaccines

Recent advances have propelled mRNA vaccines into the spotlight due to their immense therapeutic potential [71]. These vaccines operate by delivering antigen-encoding mRNA—typically via lipid nanoparticle (LNP) carriers - into human cells. Once internalized, the mRNA is translated into antigenic proteins, triggering both humoral and cellular immune responses to establish protective immunity [72,73] (Figure 3f). The production of mRNA vaccines does not require high biosafety level facilities, and the vaccines themselves are completely nontoxic [74]. Nevertheless, mRNA vaccines are highly sensitive to nucleases and have more restrictive production requirements. Sometimes they need to be transported and stored under ultra-low temperature conditions [75]. Therefore, the stability of mRNA and in vivo delivery systems require further exploration and optimization.

**Figure 3 toxins-17-00222-f003:**
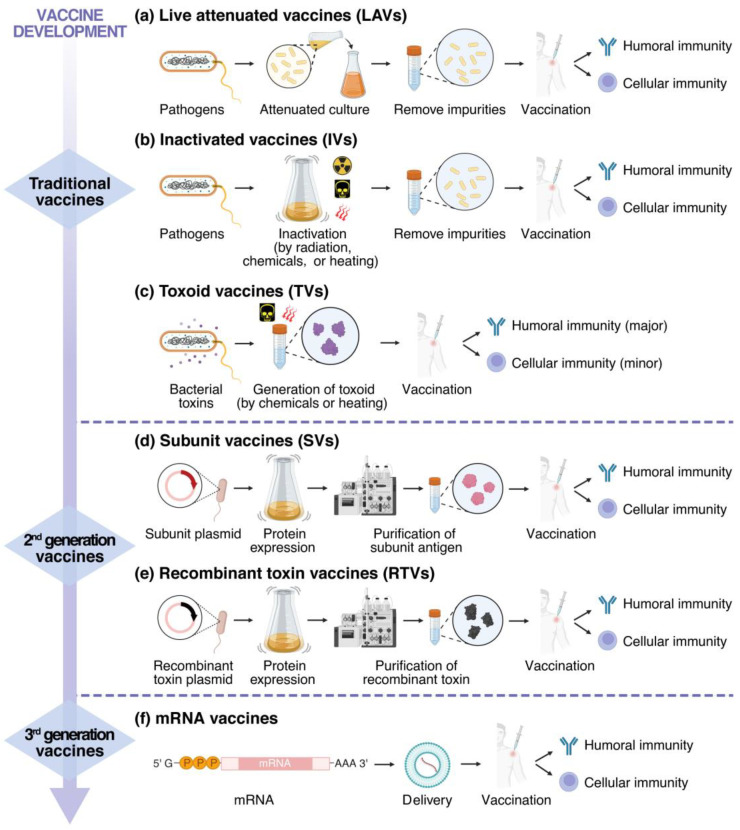
Different strategies for vaccine development. Traditional vaccines include live attenuated vaccines (**a**), inactivated vaccines (**b**), and toxoid vaccines (**c**). Second-generation vaccines include subunit vaccines (**d**) and recombinant toxin vaccines (**e**). Third-generation vaccines include the mRNA vaccine (**f**), which uses mRNA as the basic material for constructing vaccines, representing the latest progress in contemporary vaccine development [58]. (Originally, created with BioRender.com, accessed on 1 October 2024).

## 4. *C. difficile* Vaccines Studies Based on TcdA and TcdB

### 4.1. Vaccines in Clinical Trials

Currently, *C. difficile* vaccine candidates that target TcdA and TcdB have been developed by Sanofi, Pfizer, and Valneva. They have been tested in clinical trials (Table 1).

#### 4.1.1. Cdiffense^TM^ Vaccine

Cdiffense^TM^, a bivalent toxoid vaccine developed by Sanofi. It is composed of purified TcdA and TcdB from the high-toxin strain VP110463 using traditional methods. This toxoid vaccine is detoxified with formalin, using aluminum hydroxide as the adjuvant. According to publicly available information, multiple Phase I clinical trials (NCT00214461, NCT00127803, and NCT 00772954) have shown that in 50 healthy adult volunteers (aged 18–55 years) and 48 elderly volunteers (aged ≥ 65 years), this vaccine had no safety concerns [76]. Phase II clinical trials (NCT01230957, NCT00772343) identified the optimal regimen: a 100 µg antigen dose with aluminum hydroxide administered at days 0, 7, and 30 elicited sustained immune responses through day 180 in 661 participants (aged 40–75 years), warranting further clinical evaluation. [77]. However, the interim report of a Phase III clinical trial (NCT01887912) revealed insufficient efficacy in preventing primary CDI, leading to trial termination [31].

#### 4.1.2. PF-06425090 Vaccine

PF-06425090 vaccine is a recombinant toxin vaccine developed by Pfizer. The key residues in the GTDs of TcdA and TcdB have been mutated (TcdA: D285A/D287A; TcdB: D286A/D288A), resulting in reduced toxicity. These modified toxins are then expressed in a non-spore-forming *C. difficile* strain lacking toxin genes [78]. The produced toxins are further detoxified by chemicals and use aluminum hydroxide as an adjuvant. In a Phase I clinical trial (NCT01706367), this vaccine was found to be safe and highly immunogenic in healthy adults (aged 50–85 years). The toxoid-only group’s response was higher than that in the aluminum hydroxide-containing groups. However, there are some common adverse reactions following vaccination commonly seen in patients aged 50–64 years, such as pain, headache, and fatigue at the vaccination sites [79]. A Phase II clinical trial (NCT02561195) confirmed safety, tolerability, and potent immune responses in older adults (aged 65–85 years), particularly with monthly 200 μg dosing [80]. A Phase III clinical trial (NCT03090191) was initiated in early 2017. However, in March 2022, it was announced that this vaccine failed to achieve the primary endpoint of infection prevention, although its potential in the secondary endpoints, such as reducing disease duration and improving disease severity, had been observed [31,81].

#### 4.1.3. VLA84 Vaccine

The VLA84 subunit vaccine exclusively incorporates the DRBDs of TcdA and TcdB. The production of this vaccine does not require the culturing of *C. difficile* and can be prepared using a heterologous expression system [82]. A Phase I clinical trial (NCT01296386) demonstrated its safety, tolerability, and high immunogenicity across adults (aged 18–65 years) and elderly cohorts (aged ≥ 65 years), with adjuvant use showing negligible impact on immune potency [82]. In a Phase II clinical trial (NCT02316470), in 250 adult volunteers (aged 50–64 years) and 250 elderly volunteers (aged ≥ 65 years), the neutralizing antibodies produced by this vaccine were able to block the activity of TcdA and TcdB [31].

**Table 1 toxins-17-00222-t001:** Current status of *C. difficile* vaccines in clinical trials.

Vaccine Candidate	R&D Company	Status	Vaccine Type	Contents	Results
Cdiffense	Sanofi	Phase Ⅲ (terminated)	TV	Purified TcdA and TcdB from natural *C. difficile* strain VP110463; Detoxified by formaldehyde; Aluminum hydroxide as an adjuvant.	Could generate an immune response against TcdA and TcdB; Could not achieve the expected goal of preventing primary CDI [31].
PF-06425090	Pfizer	Phase Ⅲ	RTV	Recombinant TcdA and TcdB with loss-of-function mutations; Detoxified by formaldehyde; Aluminum hydroxide as an adjuvant.	Failed to reach the primary endpoint of infection prevention [31]; Could reach secondary endpoints, including reducing disease duration and improving disease severity [31].
VLA84	Valneva	Phase Ⅱ	SV	Recombinant protein subunits containing the DRBDs of TcdA and TcdB; Prepared in a heterologous expression system without culturing *C. difficile*.	The produced neutralizing antibodies could block the activity of TcdA and TcdB.

### 4.2. Preclinical Studies

In addition to the *C. difficile* vaccines based on TcdA and TcdB in clinical trials, there are some promising vaccines in the preclinical stage. They have been demonstrated to have robust protective efficacy in animal models. Recently, a multivalent mRNA vaccine (mRNA-LNP) against CDI has been developed [83]. The initial vaccine encapsulates the mRNA into LNPs that encode the CROPs and DRBD of TcdA and TcdB, and a metalloprotease virulence factor Pro-Pro endopeptidase 1 (PPEP-1/Zmp1). In murine models, this vaccine confers protection against lethal CDI in both primary infection and recurrent infection scenarios, exhibiting approximately 2-to-4-fold higher IgG responses compared to RTVs [83]. Furthermore, an extra *C. difficile* spore coat antigen, CdeM, has been added to mRNA-LNPs, and the results indicate that this approach can be more efficient to protect against disease progression, reduce colonization, and promote the decolonization of toxigenic *C. difficile* from the gastrointestinal tract. Importantly, the multivalent vaccine elicits potent systemic and mucosal antigen-specific humoral responses while maintaining intestinal microbiome homeostasis, with additional evidence of cellular immunity activation in animal studies [83]. This study can target the important bacterial pathogen and lead the new way to future vaccine development [84].

## 5. Preclinical Studies Based on Other Antigens

While current vaccine candidates in clinical trials primarily utilize TcdA and TcdB as antigens, alternative *C. difficile* antigens show promise as immunogenic targets. These include binary toxin (CDT), cell surface antigens [85,86,87], spore coat antigens [32], and flagella [86]. These macromolecules are unique to *C. difficile* and are the major contributors to CDI initiation and recurrence. Therefore, these antigens provide significant value and a novel route for the development of new vaccines (Table 2).

### 5.1. Using CDT as Antigen

Structurally and functionally distinct from TcdA/TcdB, CDT demonstrates toxin-potentiating effects, leading to more severe pathological reactions [88,89,90]. Research has shown that a quadrivalent vaccine combining attenuated TcdA, TcdB, and CDT components significantly enhances protection against hypervirulent NAP1 strains in both hamster and rhesus monkey models [91]. In another study, a chimeric protein fused by CDTb and the DRBDs of TcdA and TcdB variants from the 630 NAP1 strain has been developed [92]. Immunization induces cross-neutralizing antibodies in mice and hamsters, conferring resistance to *C. difficile* spore challenges while also stimulating human-compatible polyclonal antitoxin IgG production in transgenic cattle.

### 5.2. Antigens Involved in C. difficile Early Colonization

The surface layer proteins (SLPs) of *C. difficile* not only facilitate bacterial adhesion to intestinal epithelial cells but also serve as antigens to induce an immune response in CDI patients [93]. As a precursor protein of SLPs, SlpA is cleaved by the cell wall cysteine protease Cwp84 during secretion and is decomposed into a high-molecular-weight SLP (HMW-SLP) and a low-molecular-weight SLP (LMW-SLP) (Figure 4). These components assemble into a high-affinity heterodimeric SLP, representing the basic subunit of the S-layer [94,95]. Clinical relevance is evidenced by SlpA-specific serum antibodies detected in CDI patients. Adjuvanted with cholera toxin subunits, SlpA vaccines provoke both local and systemic humoral immune responses in multiple animal models [96].

In addition, the flagella play a crucial role in promoting *C. difficile* adhesion, invasion, and colonization by increasing the interaction between the pathogens and the epithelial mucosa [97]. The flagella of *C. difficile* are mainly composed by two proteins, FliC and FliD, both of which are involved in *C. difficile* adhesion (Figure 4) [98]. A fusion protein vaccine containing FliC and FliD (FliCD) has been developed [99]. It can induce effective IgG and IgA responses and reduce the levels of *C. difficile* spores and toxins in mice feces. Additionally, four promising epitopes of FliC and FliD have been identified using in silico and PEPSCAN procedure [100], creating opportunities for epitope-targeted vaccine design.

### 5.3. Phosphorylated Polysaccharides

The specific phosphorylated polysaccharides (PSs) exposed in the *C. difficile* cell wall, named PS-I, PS-II, and PS-III, are potential targets for vaccine development [101]. In particular, PS-II (hexasaccharide phosphate) has been identified in all *C. difficile* strains [102]. Preclinical testing of synthetic oligosaccharide-based vaccines has demonstrated both PS-specific antibody production and protection against multiple *C. difficile* strains in murine models [103]. It has been found that the serum of foals inoculated with 500 μg PS-II showed increased IgM responses [104]. Notably, serum IgM levels peaked one week post-secondary vaccination without significant adverse effects, supporting further investigation of carbohydrate-based strategies for CDI prevention.

### 5.4. Spore Coat Proteins

*C. difficile* spores can persist in the host and transmit infection [105]. Systematic analysis of *C. difficile* strain 630 has identified 54 spore-associated proteins categorized by functional localization [106]. These proteins are classified into four categories based on their possible functions and localizations: (1) exosporium proteins, such as BclA1-3, CdeC, and CdeM; (2) related to the morphogenesis of spore shells, such as CotA; (3) related to spore resistance, such as CotG; and (4) related to germination, such as cortex-lytic enzyme SleC [106]. Since these spore-associated proteins are important in the life cycle and pathogenicity of *C. difficile*, they may serve as potential targets for vaccine development.

Conserved across *C. difficile* lineages, BclA family proteins demonstrate varied vaccination potential [107]. BclA1 has been revealed to play a critical role in the initial stages of infection (e.g., pre-spore germination) in mice and hamsters [108]. BclA1 holds potential for vaccination, although intraperitoneal injection of recombinant BclA1 in mice did not confer protective immunity [106]. BclA2 (C-terminal domain) has been found to induce a specific humoral immune response in nasally immunized mice. However, this immune response cannot provide effective protection [109]. In another study, in the presence of cholera toxin, animals were immunized with KLH (keyhole limpet hemocyanin)-BclA3 glycopeptides and then infected with *C. difficile* R20291 spores. The results indicate that specific antibodies are raised; however, immunization cannot provide protection against acute or recurrent diseases [110]. CdeC and CdeM, as cysteine-rich proteins, are essential for the assembly of *C. difficile* exosporium [111]. These proteins have shown immunogenicity in animal models when administered via the intraperitoneal route, and they are protective for mice and golden Syrian hamsters against *C. difficile* [106]. Spore shell proteins, such as CotA, CotE, CdeC, and CdeH, have been identified as potential vaccine targets [112,113]. Intraperitoneal vaccination with recombinant CotA has been shown to elicit a significant IgG response after the third administration. CotA-specific IgG can approach the inside of exospores, contributing to protective immunity. Additionally, five immunogenic proteins (including spore coat proteins CotE, CotA, and CotCB; a cytosolic methyltransferase; and an exosporium protein CdeC) have been identified as potential candidate epitopes located on the outer layer of *C. difficile* spores. CotE has been selected to formulate a multivalent chimeric protein vaccine together with SlpA and FliC, which has been demonstrated to trigger protective immune responses effectively [86].

**Table 2 toxins-17-00222-t002:** *C. difficile* vaccine candidates in preclinical studies.

Antigen Types	Antigens	Results
Toxin	TcdA and TcdB	The multivalent mRNA vaccine composed of TcdA, TcdB, and CdeM, can induce strong systemic and mucosal antigen-specific humoral and cellular immune responses in mice and hamsters [83]. Does not damage the gut microbiota [83].
	CDT	The vaccine including DRBDs of TcdA, TcdB, and CDT can enhance the efficacy against the *C. difficile* NAP1 strain in mice and hamsters [92].
Surface antigens	SlpA	*C. difficile* SlpA is immunogenic [96]. Vaccination with SlpA induced partial protection in hamsters [96]. Vaccination with SlpA led to a decrease *C. difficile* gut colonization in mice [96]. Immunogenicity varies based on adjuvants [32,96].
	Cwp84	Cwp84 is conserved and highly immunogenic but is susceptible to degradation in the gut [114].
Flagella	FliC	Oral immunization with FliC-loaded beads can induce a mucosal immune response in hamsters [115]. FliC can be used as an adjuvant in the mucosal vaccination strategy [116].
	FliD	FliD plays a role in *C. difficile* adherence to mucus and epithelial cells [117]. The rectal route is the most efficient in mice vaccination [114].
Spore coat antigens	BclA1, BclA2, and BclA3	Intraperitoneal injection of BclA1 has no protective effect in mice [20]. The C-terminal domain of BclA2 can induce IgG responses but cannot mitigate CDI symptoms in mice [109]. BclA3 immunization can stimulate specific antibodies but cannot protect against acute or recurrent disease in mice and rabbits [110].
	CdeC and CdeM	Can protect mice and golden Syrian hamsters against *C. difficile* [106].
	CotA	Intraperitoneal vaccination of CotA can result in a significant IgG response [20,106] and prevent the death of mice [32,106].
Phosphorylated polysaccharides	PS-I, PS-II, and PS-III	Serum IgM responses were induced by PS-II vaccination in foals with no significant adverse reactions [104]. An appropriate carrier is required [118].

## 6. Discussion and Future Directions

Advances in pathogenic mechanism understanding and immunology, combined with biotechnological innovations, form the foundation for next-generation vaccine development. Despite decades of investigation, no FDA-approved *C. difficile* vaccine currently exists on the market. With the increasing threat posed by CDI, the development of effective vaccines to prevent the outbreak of epidemics is urgent. The virulence factors of *C. difficile* are classified into toxins (including TcdA, TcdB, and CDT) and non-toxin pathogenic factors (such as spore proteins, adhesion factors, and flagella). The central role of TcdA- and TcdB-induced tissue damage in CDI pathogenesis has historically guided vaccine design and antibody therapies toward these toxins. Although toxin-based vaccines may potentially prevent CDI, they do not address the colonization and spore germination of *C. difficile*, which can lead to the persistence of asymptomatic carriers and recurrent infections. As a result, research has increasingly focused on targeting other antigens, including cell wall proteins, flagellar proteins, and spore surface proteins, all of which play a key role in early bacterial colonization. These new antigens offer several advantages, and some vaccine candidates have shown promising preclinical results. However, further studies are needed to overcome the current limitations and to determine the most effective epitopes. Additionally, vaccines that combine multiple antigens, such as various toxin domains, may provide a more comprehensive approach to prevent CDI. However, the safety and efficacy issues of many vaccine candidates often arise during clinical practice, necessitating a continued focus on optimizing vaccine compositions, exploring alternative adjuvants, and improving delivery systems. With these advancements, it is hoped that CDI can be effectively controlled, or even eradicated, in the near future.

## Figures and Tables

**Figure 1 toxins-17-00222-f001:**
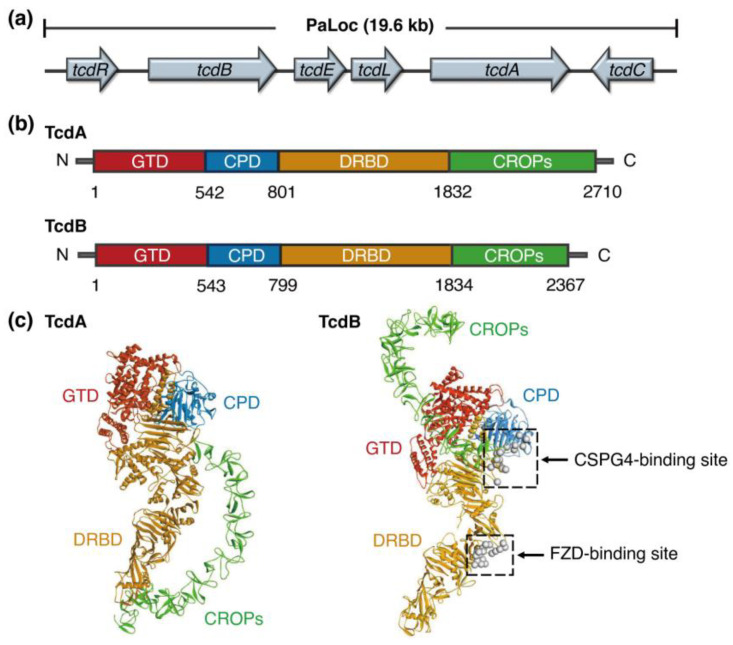
TcdA and TcdB are major virulence factors of pathogenic *C. difficile* strains. (**a**) The pathogenicity locus (PaLoc) from a representative *C. difficile* strain. (**b**) A scheme of the functional domains of TcdA and TcdB (numbers indicate residue sites). N, N-terminus; C, C-terminus; GTD, glucosyltransferase domain; CPD, cysteine protease domain; DRBD, a mixed membrane translocation delivery and receptor-binding domain; CROPs, combined repetitive oligopeptide domain. (**c**) Structures of TcdA (PDB ID: 7POG) and TcdB (PDB ID: 6OQ5). The receptor binding sites on TcdB are indicated. FZD, frizzled receptor; CSPG4, chondroitin sulfate proteoglycan 4.

**Figure 2 toxins-17-00222-f002:**
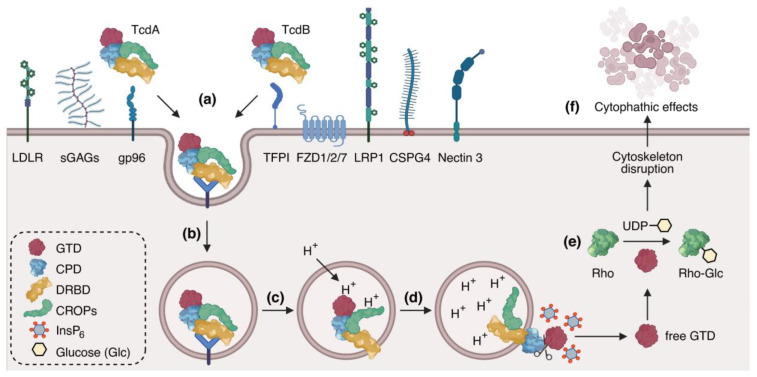
Intoxication mechanism of TcdA and TcdB. (**a**) Toxins bind their specific cellular receptors. (**b**) Toxins enter endosomes through endocytosis. (**c**) A reduction in endosomal pH induces conformational changes of DRDB and the membrane translocation of GTD and CPD. (**d**) In the presence of InsP_6_, CPD auto-cleavages and releases GTD. (**e**) The free GTD targets and glycosylates small GTPases using UDP-glucose as a sugar donor. (**f**) The actin cytoskeleton is disrupted, resulting in the cytophathic effect, such as cell rounding and apoptosis. (Originally created with BioRender.com, accessed on 1 October 2024).

**Figure 4 toxins-17-00222-f004:**
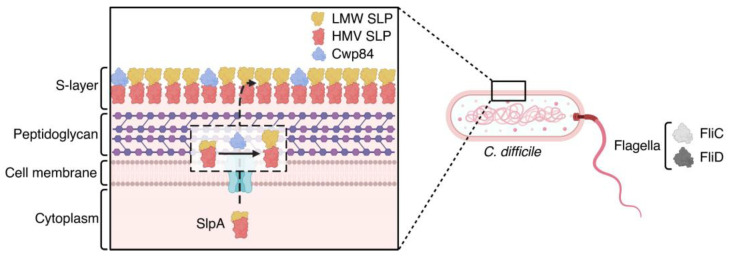
*C. difficile* surface antigens. The *C. difficile* flagella consist of FliC and FliD. SlpA is cleaved by Cwp84 (represented in purple) during secretion and decomposed into a high-molecular-weight SLP (HMW-SLP) (represented in red) and a low-molecular-weight SLP (LMW-SLP) (represented in yellow) [94,95]. (Originally created with BioRender.com, accessed on 1 October 2024).

## Data Availability

No new data were created or analyzed in this study.

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
