# Peer review of "Against Clostridioides difficile Infection: An Update on Vaccine Development"

_toxins, 2025, doi:10.3390/toxins17050222_

Round 1
Reviewer 1 Report
Comments and Suggestions for Authors
The review article entitled “Against Clostridiodes difficile infection: An update on Vaccine development” is a comprehensive review of the current literatures available about the various kinds of vaccines as well as the putative vaccines at different stages of developmental against this bacterial disease. The authors, after introducing the topic, first summarize the biology of C. difficle toxins and conclude that TcdA & TcdB are two protein exotoxins which regulate many physiological events within host tissue leading to pathology and culminating to disease. Based on the survey of literature as cited in this review the investigators determine that passive immunization with toxin neutralizing antibodies have protected host against Clostridiodes difficile infection and therefore TcdA & TcdB could be candidate vaccine antigens for this disease. The authors further talk about traditional, first, second and third generation vaccines against other bacterial diseases. Additionally, they also provide a summary of the various putative vaccines against C. difficile which are at different stages of clinical trials. The authors conclude that understanding of the pathogenic mechanisms and immune responses of the host against infectious diseases are of paramount importance for design and development of a new vaccines against C. difficile.
Author Response
Reviewer #1:
The review article entitled “Against Clostridiodes difficile infection: An update on Vaccine development” is a comprehensive review of the current literatures available about the various kinds of vaccines as well as the putative vaccines at different stages of developmental against this bacterial disease. The authors, after introducing the topic, first summarize the biology of C. difficle toxins and conclude that TcdA & TcdB are two protein exotoxins which regulate many physiological events within host tissue leading to pathology and culminating to disease. Based on the survey of literature as cited in this review the investigators determine that passive immunization with toxin neutralizing antibodies have protected host against Clostridiodes difficile infection and therefore TcdA & TcdB could be candidate vaccine antigens for this disease. The authors further talk about traditional, first, second and third generation vaccines against other bacterial diseases. Additionally, they also provide a summary of the various putative vaccines against C. difficile which are at different stages of clinical trials. The authors conclude that understanding of the pathogenic mechanisms and immune responses of the host against infectious diseases are of paramount importance for design and development of a new vaccines against C. difficile.
Response: Thank you for your careful review of our manuscript. We have conducted a comprehensive language edit and made necessary formatting adjustments by native English speakers to ensure that the manuscript meets the journal's standards.
Reviewer 2 Report
Comments and Suggestions for Authors
This review manuscript summarized the vaccines research and clinical studies so far, it is very informative for the researchers in this area. It is very suitable for the people who first starts the research in this area to understand and pick up the pace for the studies in this area! Great match for this journal.
I only have one suggestion or question, which I hope the authors can answer me:
Since the TcdA and TcdB has so many different types and subtypes, will the vaccine development be harder? And if the TcdAB are easier to change in different ribotypes of C. diff, is this also an advantage for the mRNA vaccine?
Author Response
Reviewer #2:
Q1. This review manuscript summarized the vaccines research and clinical studies so far, it is very informative for the researchers in this area. It is very suitable for the people who first starts the research in this area to understand and pick up the pace for the studies in this area! Great match for this journal. I only have one suggestion or question, which I hope the authors can answer me: Since the TcdA and TcdB has so many different types and subtypes, will the vaccine development be harder?
Response: This is a good point. TcdA and TcdB exhibit multiple subtypes (TcdA1-A7 and TcdB1-B12) with diverse pathological properties, particularly in receptor specificity. Despite this functional variation, these subtypes maintain high sequence similarity - a likely consequence of their large size (>2,000 residues; TcdA ~306 kDa, TcdB ~270 kDa). The sequence alignment and Neighbor-joining phylogenetic analysis revealed that: TcdA subtypes A1-A6 share >97% identity; a distinct TcdA7 subtype maintains ~85% identity with others; tcdB subtypes show greater dispersion but still maintain >85.3% identity [PLoS Pathog. 2020 Dec 28;16(12):e1009181, PLoS Pathog. 2021 Jan 28;17(1):e1009197]. This high conservation suggests numerous surface epitopes suitable for vaccine development that may confer protection across multiple subtypes. On the other hand, epidemiological data from clinical isolates indicate three dominant subtypes account for most infections: TcdA1 (~63.2%), TcdB1 (~63.2%), and TcdB2 (~6.8%). This prevalence pattern strongly supports focusing vaccine development efforts on these three antigenic subtypes. We hope the reviewer's concern can be addressed.
Q2. And if the TcdAB are easier to change in different ribotypes of C. diff, is this also an advantage for the mRNA vaccine?
Response: The effectiveness of toxin-targeting vaccines may be impacted by the variability of TcdA/B toxins across different C. difficile ribotypes. While the modular nature of mRNA vaccine technology offers potential for rapid adaptation to emerging ribotypes. The current mRNA vaccine research has primarily focused on membrane-associated antigens (e.g., SARS-CoV-2 spike proteins), with limited exploration of applications against secreted toxins. Several key challenges, such as durability of immune responses and optimal antigen selection strategies, remain to be explored. We hope the reviewer's concern can be addressed.
Reviewer 3 Report
Comments and Suggestions for Authors
difficile causes severe diarrhea and colitis, especially in people taking antibiotics or in healthcare settings. Recurrence is common even after treatment. A vaccine could prevent primary infections and reduce healthcare-associated disease. This manuscript is very interesting and hilights importance of new treatment for recurente colitis . Authors should add more information about FMT as a potetntial vaccin. Thank you for the opportunity to review this paper.
Author Response
Reviewer #3:
Q1. difficile causes severe diarrhea and colitis, especially in people taking antibiotics or in healthcare settings. Recurrence is common even after treatment. A vaccine could prevent primary infections and reduce healthcare-associated disease. This manuscript is very interesting and hilights importance of new treatment for recurente colitis. Authors should add more information about FMT as a potetntial vaccin. Thank you for the opportunity to review this paper.
Response: We appreciate this valuable suggestion. Accordingly, we have expanded our discussion of fecal microbiota transplantation (FMT) in the revised manuscript (Introduction section, highlighted in bright color). Current evidence shows that FMT is highly effective for treating recurrent CDI, but its application remains primarily therapeutic rather than preventive. FMT works by transferring healthy donor microbiota to restore gut microbial balance, which is fundamentally different from vaccination. Vaccines induce active immunity, whereas FMT provides passive microbial reconstitution. Additionally, FMT carries risks (e.g., potential pathogen transmission) that may outweigh benefits in asymptomatic individuals, unlike vaccines with well-defined safety profiles. Patient acceptance may also be limited due to psychological reservations about the procedure. While FMT has revolutionized CDI treatment, its role in prevention requires further investigation. We hope this clarification addresses the reviewer’s concern.